# NeuroNet: Fast and Robust Reproduction of Multiple Brain Image Segmentation Pipelines

**Martin Rajchl**[*]
Depts. of Computing and Medicine
Imperial College London

**Nick Pawlowski**
Dept. of Computing
Imperial College London

**Daniel Rueckert**
Dept. of Computing
Imperial College London

**Paul M. Matthews**
Dept. of Medicine
Imperial College London

**Ben Glocker**
Dept. of Computing
Imperial College London

## Abstract

NeuroNet is a deep convolutional neural network mimicking multiple popular and state-of-the-art brain segmentation tools including FSL, SPM, and MALPEM. The network is trained on 5,000 T1-weighted brain MRI scans from the UK Biobank Imaging Study that have been automatically segmented into brain tissue and cortical and sub-cortical structures using the standard neuroimaging pipelines. Training a single model from these complementary and partially overlapping label maps yields a new powerful "all-in-one", multi-output segmentation tool. The processing time for a single subject is reduced by an order of magnitude compared to running each individual software package. We demonstrate very good reproducibility of the original outputs while increasing robustness to variations in the input data. We believe NeuroNet could be an important tool in large-scale population imaging studies and serve as a new standard in neuroscience by reducing the risk of introducing bias when choosing a specific software package.

## 1 Introduction

Accurate and robust structural segmentation of the brain is a key component in neuroimaging research. Semantic segmentation and, thus, the identification of cortical and subcortical structures allows quantification of anatomical variation (*e.g.*, hippocampal volume, gray matter thickness, white matter loss, *etc.*) and relation of brain function and connectivity to meaningful spatial locations. Well-established segmentation tools, such as FSL [1], SPM [2] and others [3], have been developed and frequently employed over the last decade. These tools exhibit different strengths and weaknesses [4, 5] and neuroscientists are left with an agony of choice knowing that different tools might introduce different biases [6] (*c.f.* the subcortical GM segments in Figure 1). This may negatively impact findings and weaken any drawn conclusions.

Further, traditional pipelines require extensive pre-processing of the input images to improve the initial conditions for the subsequent segmentation method. This can include spatial normalisation via registration to an atlas (*e.g.* MNI152[2]), correction of the bias field [8] and employing brain stripping methods [9], further exacerbating the computational burden required to process a subject (*c.f.* running all packages to obtain the outputs shown in Fig. 1 requires several hours per scan).

---

[*]m.rajchl@imperial.ac.uk
[2]http://www.bic.mni.mcgill.ca/ServicesAtlases/ICBM152NLin6

1st Conference on Medical Imaging with Deep Learning (MIDL 2018), Amsterdam, The Netherlands.

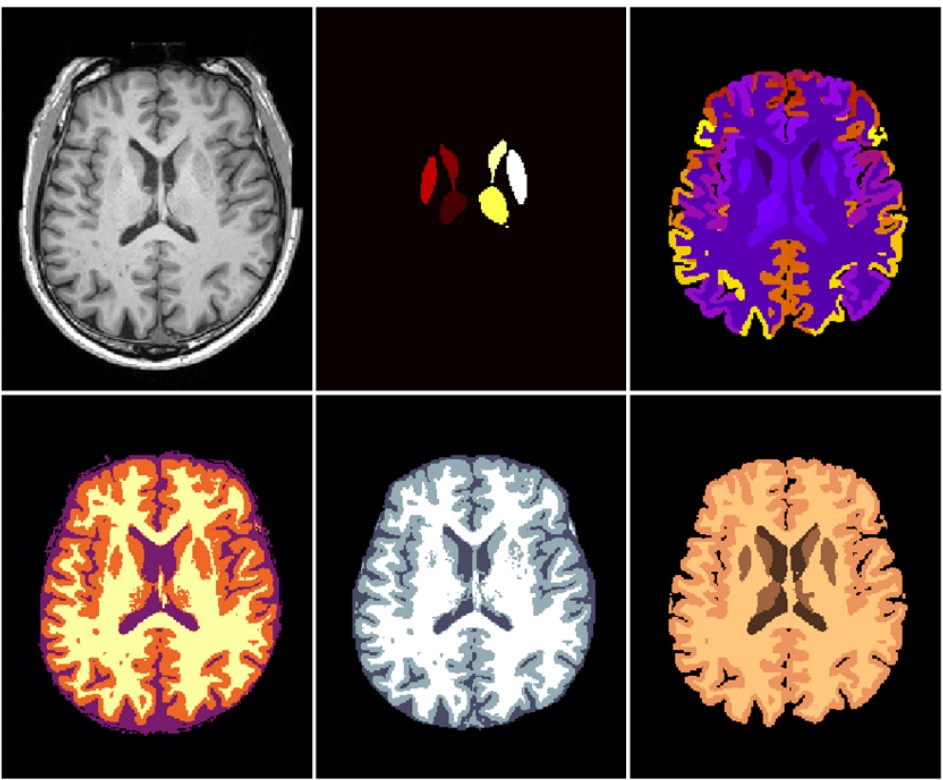

Figure 1: Random exemplary UK Biobank case processed with FSL [1], SPM [2] and MALP-EM [3] after pre-processing according to [7]: (from top left to bottom right): Raw T1w MR image, FSL First, MALP-EM, SPM tissue, FSL fast, MALP-EM tissue segmentations.

With ambitious imaging studies on unprecedented scales [10], the computational burden of current image processing pipelines is prohibitive. However, recent advances in machine learning, particularly fully convolutional neural networks (CNNs) allow for fast inference on imaging data, addressing this impediment. Further, CNNs currently rank highest on accuracy collation studies [11–13] for most segmentation tasks on natural and medical images [14–16] alike.

In order to process neuroimaging data on such large scales, we require tools that closely reproduce outputs of well established packages in a more robust (*c.f.* Figure 2) and efficient manner as basis for large-scale analyses.

## 1.1 Contributions

To this end, we developed NeuroNet, a comprehensive brain image segmentation tool based on a novel multi-output CNN architecture which has been trained to reproduce simultaneously the output of multiple state-of-the-art neuroimaging tools. By learning jointly from complementary and partially overlapping tissue label maps, NeuroNet produces not only highly accurate structural segmentations but can also reduce the number failure cases compared to current methods (*c.f.* Figure 2).

Neuronet learns to produce outputs from raw images, avoiding the need for typical pre-processing tools (such as bias correction or brain-stripping), thus further reducing a source of errors and the need for adjusting additional hyper-parameters.

A key aspect in NeuroNet is the training from a large imaging database, the UK Biobank, which is one of the world's largest ongoing population imaging studies. We currently utilize imaging data from more than 5,000 subjects where each has been automatically segmented using three different state-of-the-art tools, FSL, SPM and MALP-EM generating five outputs in total. The automatically generated label maps from the three tools serve as training data for NeuroNet which aims to mimic

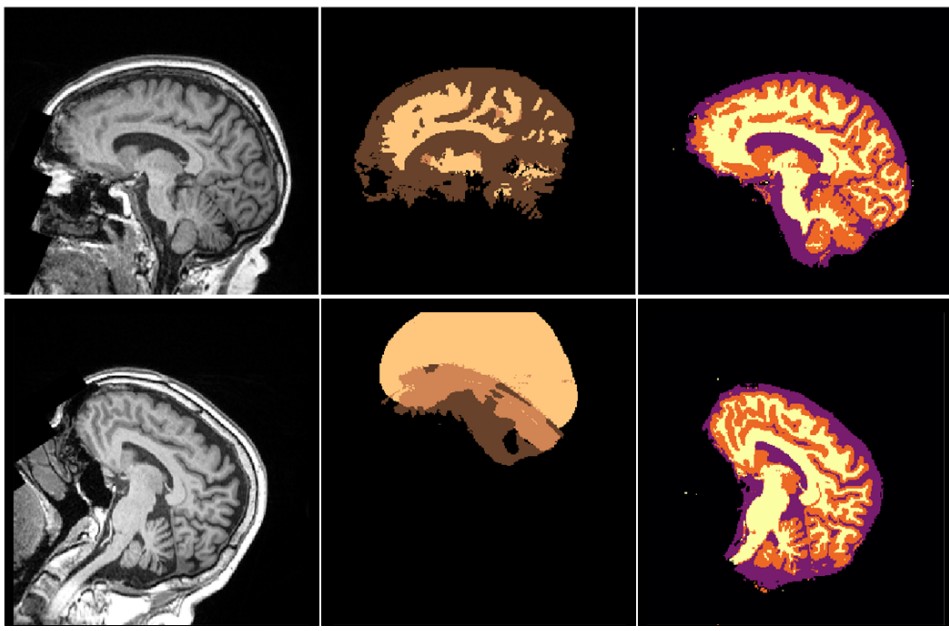

Figure 2: Two cases of a T1w MR (left) where processing with SPM failed (middle) while high quality predictions are obtained with our deep learning based NeuroNet (right).

the performance of each tool by learning the complex non-linear associations between input image data and output tissue class label maps.

The label maps used for training that are produced by FSL, SPM and MALP-EM differ in the number of structures and their granularity of segmentation, and result in complementary but also partially overlapping semantic label sets. For example, all three tools generate estimates for white matter (WM), gray matter (GM) and cerebro-spinal fluid (CSF), while in addition FSL and MALP-EM each produce detailed maps of substructures with 16 labels in case of FSL and 138 in case of MALP-EM. For structures that are present in the output of all three tools such as WM, GM and CSF, we obtain three different estimates, one from each tool, which are all used as gold-standard reference. An intriguing characteristic of our multi-output approach arises from the fact that during training NeuroNet is presented with this variation (or uncertainty) in the reference and is forced to learn a consensus prediction. Theoretically, this should lead to a more robust estimation of the underlying true labeling where errors and biases introduced by each individual tool are averaged out as part of the training process. Additionally, learning jointly from hierarchical sets of class labels has the potential to increase the overall accuracy based on theory derived from multi-task learning.

## 2 Methods

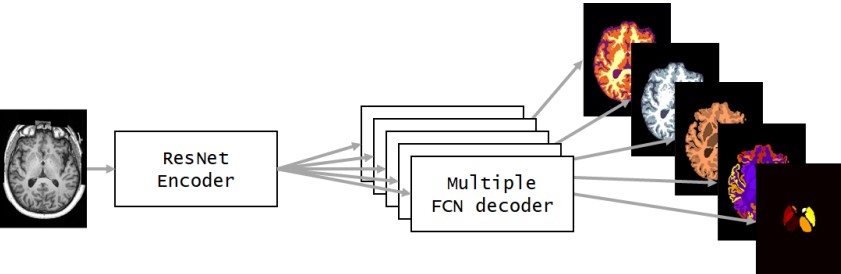

Figure 3: Proposed multi-output architecture, NeuroNet. If the number of outputs is one, it reduces itself to an updated FCN architecture [17] with a ResNet encoder [18] as in [16].

## 2.1 Network Architecture

For the NeuroNet architecture, we employ state-of-the art elements of well-studied networks and combine them to facilitate multi-objective learning of image segmentations. We note, that all employed network operations were implemented in 3D to learn features in all dimensions of the input space.

**Input & Output Spaces:** Given a training database of size $N$ containing images $x = \{x_1, ..., x_N\}$ and corresponding $k$ output segmentations $y^k = \{y_1^k, ..., y_N^k\}$ coming from different tools, we aim to predict $\hat{y}^k$ with a CNN with parameters $\Theta$.

**Feature encoding:** After an initial convolution, we extract features with $N_{units} = 2$ updated residual units $U_i^{S_j} = \{U_1^{S_j}, ..., U_{N_{units}}^{S_j}\}$ according to [18] on each of $N_{scales} = 4$ resolution scales $S_j = \{S_1, ..., S_{N_{scales}}\}$, employing a leaky ReLu (leakiness = 0.1) activation function as non-linearity [19] with preceding batch normalisation. At each $S$ we downsample the feature tensor via strided convolutions [20] in the first residual unit $U_1^{S_j}$, where the $strides_j = \{1, 2, 2, 2\}$ operate in each spatial dimension. Lastly, we define a fixed number of filters for all convolutions in $U^{S_j}$, doubling the number at each scale: 16, 32, 64 and 128.

**Multi-decoder architecture:** Fully convolutional networks [15, 21, 17] for image segmentation typically aim to reconstruct a prediction tensor in the same shape as the input image. Due to memory limitations, we aim to allocate as many parameters on the encoder side and keep the decoder part as light as possible. For this purpose, we base our multi-decoder architecture on FCN upscore operations [17]: To reconstruct a prediction at each resolution scale $S_{j-1}$, we linearly upsample the prediction at $S_j$ and add a skip connection from the output of the last residual unit $U_{N_{units}}^{S_j}$ of the encoder. The output of the last residual unit $U_{N_{units}}^{S_0}$ at decoder $k$ serves as output of the network. A prediction $\hat{y}^k$ can be obtained via $softmax$. We repeat this process for each of the $k$ decoders.

**Multi-task Loss:** NeuroNet's multi-task learning problem is addressed similarly to [22], with the objective to minimize a total loss $L_{Total}$ as the weighted sum of the $k$ individual losses $L_{Ind.}$:

$$L_{Total} = \sum_{i=1}^{k} \lambda_i \, L_{Ind.}(\hat{y}^i, y^i) \,, \tag{1}$$

where $\lambda$ is a weighting parameter. We simplify the problem by employing the same categorical cross-entropy loss for all prediction outputs $\hat{y}^k$ at voxel locations $v$,

$$L_{Ind.}(\hat{y}, y) = - \sum_{v} \hat{y}(v) \log y(v) \,, \tag{2}$$

and weight them equally with $\lambda = \frac{1}{k}$, resulting in an average cross-entropy loss.

## 2.2 Training

We train all compared networks using the Adam optimiser [23] with a constant learning rate of $0.001$ and an $\epsilon$ of $10^{-5}$. Pre-processed input batches (batch_size=1) of size $128^3$ and were mixed in a queue (queue_capacity=16) and the network trained for $10^5$ steps.

## 2.3 Preprocessing & Augmentation

During training, the input volume images $x$ are normalised to zero mean and unit standard deviation using volume statistics. No other preprocessing was employed. Data augmentation is a common regimen to prevent over-fitting to the training set, where distortions and transformations are applied to the training examples to force a model to learn the artificially introduced variation and potentially generalise better. It is routine on medical image segmentation datasets, which are limited in scale, as voxel-wise (expert) manual annotations are tedious to produce. However, due to the scale of available data and training targets in form of automated tool kit outputs, we limit the augmentation to random crops of size $128^3$ to guarantee equally sized input tensors for the network.

# 3 Experiments

## 3.1 Data

The UK Biobank imaging study is an ambitious undertaking of producing a volunteer database of unprecedented scale. It aims to collect over 100,000 subjects over time, including a comprehensive imaging protocol. For this study, we downloaded the first 5000+ datasets (under application 12579), including the brain MRI entries of raw and bias-corrected T1w and T2w structural images, brain masks generated with BET [9], tissue segmentations with FSL Fast [1] and subcortical GM segmentations with FSL First. The complete documentation can be found in [7]. Additionally, we generated tissue segmentations with SPM-12 [2] and a state-of-the-art multi-atlas segmentation tool MALP-EM [3]. The outputs of MALP-EM are a full 139 label segmentation and a secondary 5 label tissue segmentation variant (see Fig. 1). The tool typically employs PINCRAM [24] for skull stripping, which was replaced with the provided BET brain segmentation mask, as it was consistently under-segmenting on these data. After a completeness check of all entries, we split the 5,723 available subjects into 5,000 training, 10 validation and 713 test datasets. For our experiments, the 10 validation subjects were sufficient to determine potential over-fitting to the training data.

## 3.2 Implementation

NeuroNet is implemented using DLTK [16] and TensorFlow [25] with a NIfTI image IO interface using SimpleITK [26]. As the size of the database largely exceeds available memory, we employ a shuffling queue to pre-load and augment the data online during training. For reproducibility, the complete source code, configuration files and all trained models used in these experiments will be made available online in the DLTK Model Zoo [3]. Due to its design of allocating the majority of network parameters on the encoder side (see 2.1, we can fit NeuroNet into the memory of cheap, commercially available graphics hardware.

## 3.3 Evaluation

In our experiments, we compare the performance of NeuroNet multi-output variants with that of single-output architectures. To ensure equal conditions in our experiments all training parameters (*e.g.* learning rate, training steps, *etc.*) and inputs remained unchanged and all runs initialised with the same seeds. The compared NeuroNet variants include training

- on single targets (*fsl_fast*, *fsl_first*, *spm_tissue*, *malp_em*, *malp_em_tissue*),
- NeuroNet on all targets (*nn_all*),
- NeuroNet on tissue segmentations only (*nn_tissue*).

We evaluate the accuracy of network predictions over the respective tool kit outputs in terms of mean Dice Similarity Coefficient (DSC) over all labels:

$$DSC = \frac{2\left|L_{Prediction} \cap L_{Target}\right|}{\left|L_{Prediction}\right| + \left|L_{Target}\right|} \tag{3}$$

Additionally, we summarise maximum run times of all compared methods, impacting their scalability on large-scale datasets.

# 4 Results

Numerical accuracy results on compared network types *single*, *nn_tissue* and *nn_all* can be found in Tab. 1, 2 and 3, respectively. Using the multi-output architectures *nn_tissue* and *nn_all* yielded slightly reduced mean DSC over training several *single* networks. However, all numerical accuracy results are close for all compared network types. Fig. 5 summarises Tab. 1, 2 and 3 graphically. A randomly picked test image and the corresponding *nn_all* prediction is depicted in Figure 4.

---

[3] https://github.com/DLTK/models

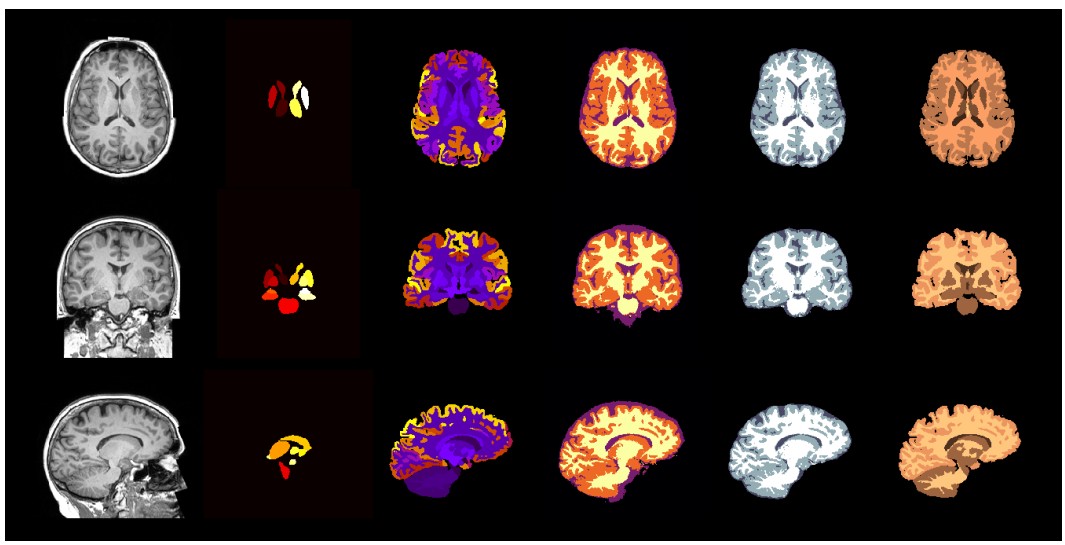

Figure 4: Random NeuroNet (*nn_all*) segmentation example. From left to right: raw T1w input image, *fsl_first*, *malp_em*, *spm_tissue*, *fsl_fast*, *malp_em_tissue*.

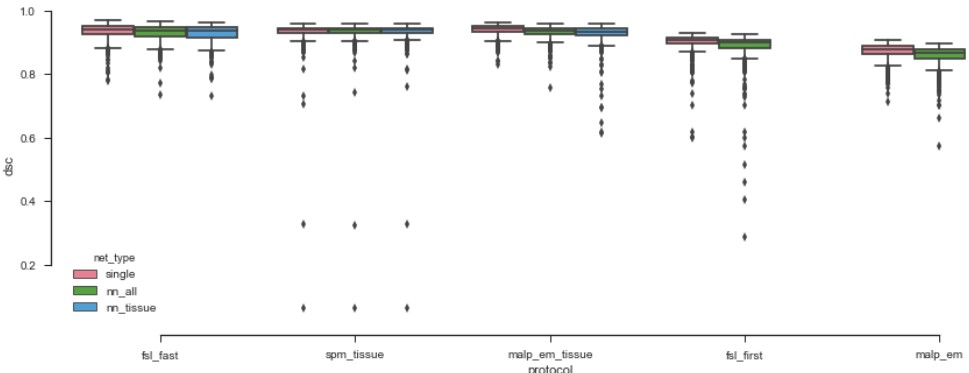

Figure 5: Comparative test accuracy results in terms of mean DSC: NeuroNet training on single (red), all (green) and tissue segmentation (blue) targets of different pipelines.

Table 1: NeuroNet Single: DSC [%] accuracy over segmentation tool kit outputs of FSL, MALP-EM & SPM

| protocol | fsl_fast | fsl_first | malp_em | malp_em_tissue | spm_tissue |
|----------|----------|-----------|---------|----------------|------------|
| mean | 93.6 | 90.2 | 87.3 | 94.0 | 93.3 |
| std | 2.5 | 3.0 | 2.6 | 1.8 | 4.5 |
| min | 78.0 | 60.0 | 71.3 | 83.2 | 6.2 |
| max | 96.9 | 93.1 | 90.8 | 96.3 | 96.0 |

Table 2: NeuroNet Tissue: DSC [%] accuracy over segmentation tool kit outputs of FSL, MALP-EM & SPM

| protocol | fsl_fast | malp_em_tissue | spm_tissue |
|----------|----------|----------------|------------|
| mean | 92.8 | 92.7 | 93.4 |
| std | 2.6 | 3.2 | 4.3 |
| min | 73.3 | 61.3 | 6.3 |
| max | 96.4 | 95.8 | 96.0 |

Table 3: NeuroNet All: DSC [%] accuracy over segmentation tool kit outputs of FSL, MALP-EM & SPM

| protocol | fsl_fast | fsl_first | malp_em | malp_em_tissue | spm_tissue |
|----------|----------|-----------|---------|----------------|------------|
| mean | 93.1 | 88.8 | 85.8 | 93.2 | 93.4 |
| std | 2.4 | 4.9 | 3.1 | 1.9 | 4.3 |
| min | 73.7 | 28.8 | 57.3 | 75.8 | 6.2 |
| max | 96.5 | 92.7 | 89.7 | 95.9 | 96.1 |

## 5   Discussion

We proposed and evaluated variants of the multi-task learning method NeuroNet, reliably reproducing outputs from well-know neuroimaging software packages. NeuroNet does not require common pre-processing steps (*i.e.* skull-stripping, additional bias-correction, etc.) and so removes a potential source of error from the pipeline. Additionally, the inference speed allows to scale known analysis pipelines to large-scale data sets, greatly enriching imaging data for further analysis.

### 5.1   Accuracy

All compared NeuroNet variants were able to produce the respective package outputs with high accuracy in terms of DSC. A qualitative comparison of reported accuracies against manual segmentations in [3] (77.2% DSC for MALP-EM) or [5] (78% DSC for SPM, 77% DSC for FSL Fast) show that NeuroNet produces highly accurate reproductions of the package outputs. When manually examining the testing outlier datasets in Figure 5 (*i.e.* with <0.6 DSC mean overlap), exclusively all can be described by a failure case of the original package output. Two exemplary failure cases are depicted in Figure 2, where due to an uncorrected head rotation SPM fails to segment the images properly. In comparison, NeuroNet was able to produce valid segmentations from learning the variations in the large training dataset.

### 5.2   Comparative Experiments

When comparing different NeuroNet variants (*i.e.* NeuroNet single, tissue and all) we find multi-task learning produces slightly reduced accuracy results over single tasks for all outputs except SPM tissue. One might think that NeuroNet all and tissue variants would have to encode more information to reconstruct all outputs, however NeuroNet all outperforms the tissue variant in all outputs in terms of mean accuracy (*c.f.* Tables 2 & 3). Considering the efforts in Section 3.3, ensuring a fair comparison between setups, it seems beneficial to add more variation in tasks. This confirms our hypothesis that learning from complementary and partially overlapping label maps is beneficial.

### 5.3   Run times

Run times for all NeuroNet variants, depending hardware is less than 90 seconds per dataset. Compared to the traditional neuroimaging packages, we measured approximate run times for FSL of 20 min per case, MALP-EM takes about 1 hour using 8 CPUs, and SPM a few minutes for tissue segmentation followed by nonlinear registration to MNI which takes another half an hour.

### 5.4   Generalisation

The proposed NeuroNet multi-task architecture is readily extendible to other learning tasks (*e.g.* regression, classification or hybrid tasks). However, when learning hybrid tasks one would have to find an appropriate set of $\lambda_i$ re-weighting the loss functions $L_{Ind}$ to $L_{Total}$, if different loss functions are employed. This poses a hyper-parameter tuning problem, thus an computational expensive undertaking. Alternatively, studies as [27] found that the $L2$ loss can be employed for classification tasks, allowing for hybrid multi-task setups of regression and classification problems. Adversarial training [28] of multiple tasks could pose another solution, avoiding the re-weighting issue.

## 5.5 Conclusions

In this paper, we proposed NeuroNet as a new, reliable scaling solution to processing neuroimaging data. The multi-task learning architecture can generalise to hybrid segmentation, classification and regression setups and so potentially facilitate interesting end-to-end learning-based solutions for neuroimage analysis problems. For the purpose of transparency and reproducibility, the entire NeuroNet code base is released publicly.

### Acknowledgements

This research has been conducted using the UK Biobank Resource under Application Number 12579. Nick Pawlowski is supported by Microsoft Research PhD Scholarship and the EPSRC Centre for Doctoral Training in High Performance Embedded and Distributed Systems (HiPEDS, Grant Reference EP/L016796/1). Ben Glocker received funding from the European Research Council (ERC) under the European Union's Horizon 2020 research and innovation programme (grant agreement No 757173, project MIRA, ERC-2017-STG). We gratefully acknowledge the support of NVIDIA Corporation with the donation of two Titan X GPUs.

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
