# OpenReview forum: "NeuroNet: Fast and Robust Reproduction of Multiple Brain Image Segmentation Pipelines"
_MIDL.amsterdam/2018/Conference — MIDL 2018 Poster_

### Review · AnonReviewer1 · 2018-05-06
**A good starting point for an all-around neuroimaging tool, more ablation tests/insights and evaluation on different datasets would be helpful**

**Rating:** 3
**Confidence:** 3

**Review:**

The authors propose to train a single CNN to mimic several popular segmentation tools for brain MRI segmentation. Overall this paper is well-structured and easy to follow. The work is however still very early stage given the tremendous aim they set: to replace FSL, SPM and the like. For this to start being credible, much more work to demonstrate the robustness of the approach is required.

Pros:

- Relatively large-scale experiments

- The proposed method has a good reproducibility of the popular segmentation tools’ outputs (on the dataset used in the paper) and requires much less processing time.

- Code and models will be made publicly available

Cons:
- Despite the fairly large-scale experiment, the data is all coming from the UK biobank thereby limiting the insight into the generalisability of the tool.

- It’s unclear how the volume predictions are generated from a network with 128^3-voxel outputs.

- “An intriguing characteristic of our multi-output approach … is forced to learn a consensus prediction” not sure how this is achieved, since the same categorical labels across the multi-task outputs are treated independently. It would be interesting to check the inter-task variability of the overlapping labels of NeuroNet.

- Since `nn_all` and the other single target tasks have significantly different levels of complexity, in my opinion using the same set of hyper-parameters (learning rate/number of training steps) might be problematic. Is there any underfitting issue for `nn_all`?

- “NeuroNet does not require common pre-processing steps and so removes a potential source of error from the pipeline” – Because NeuroNet is trained to mimic the output of the existing tools which depends on some pre-processing steps; I feel the claim of removing the pre-processing errors is questionable.

- Lack of comparisons to other encoder-decoder architectures in the literature.

- The abstract states that the network is trained on T1-weighted MRI scans but they are actually T1w and T2w (section 3.1).


**Special Issue:**

No

---

### Review · AnonReviewer2 · 2018-05-11
**Well-executed, large scale validation of a single CNN to replace many popular brain MR image analysis tools**

**Rating:** 4
**Confidence:** 3

**Review:**

The paper describes the experiments to validate a single, multi-output 3D CNN in segmenting brain structures on a large cohort of patients from the UK Biobank. The goal is to replace several existing traditional image analysis algorithms, which, although widely used, are slow and all need to be run individually. I found the idea deceptively obvious, but I searched around a bit and as far as I can tell this is the first paper that tries to do this, so hats of to the authors.

The paper is very light on methodological novelty, the network used is a relatively straightforward extension of existing architecture. Furthermore, even though we are looking at multi-task output with differing goals, the loss function is a straightforward average cross-entropy. Luckily the experiment to validate the use of the CNN to replace the traditional tools compensate this lack of novelty. The experiments are rigorous and evaluated on a large dataset. Although we cannot check without seeing the test set results, the assertion of the authors that all failure cases of the CNN were actually failure cases of the traditional methods is an impressive result. In addition, the fast run time makes the method very attractive.

I do have some questions/suggestions for some additional experiments:

- How variable is the UK BioBank dataset in terms of different scanners, different field strenghts, etc.? Does it cover the full range in practice? Or would another test set from a different source be recommend.
- If I understood correctly, all scans were from healthy volunteers. It would be interesting how the CNN behaves on patient data.
- I think it would have been nice to compare the traditional methods and the CNN in some sort of observer experiment. For example, how often would a correction be needed with the traditional methods (or how long would correction take) and how long for the CNN method? Just to get a feel for what the benefit is in terms of providing accurate annotations which need little human intervention.

**Special Issue:**

Yes

---

### Review · AnonReviewer3 · 2018-05-13
**Very useful and obvious application of a segmentation network**

**Rating:** 4
**Confidence:** 3

**Review:**

The authors used existing established segmentation tools for brain MRI (like FSL) to create a reference standard for a large database of 5000 cases and trained a 3D network to reproduce these automated outputs. The motivation is
- the network may be a lot faster
- it might give better results in cases where the standard tools fail
- it removes need for preprocessing (like bias correction)

The paper is very well written.

Deep learning wise, there is nothing really novel about NeuroNet
  * automated annotations have been used before
  * multiple outputs and losses to drive learning have also been used before
  * architecture also uses common elements, even if the combination might be less common (and the authors could have provided a detailed figure of the architecture, although they might have opted against it due to page limit, considering it wasn't the focus of the project)

W.r.t. the multi task loss: the way it is written, they averaged the loss over all decoders (I haven't looked at the implementation to verify), but if that is the case, I would disagree with their decision to do so - since the decoders are separate, their losses should be kept separate for the decoders and only be combined for the encoder (if they have tried it, and found it to be less efficient, they should have written it down)

No data augmentation was used: given the large database, I understand the decision and I don't think it would have made a significant difference, although personally I would have still used data augmentation considering the large variability of MR scans

10 validation scans is a very low number. You cannot measure performance increases on more rare cases with such a low number of scans, and you're likely to stop training too early.

The authors made (or are going to make) the code publicly available, which is always a plus, especially for such a tool, derived from other publicly available tools.
  * however, they weren't very accurate in their specification of cheap hardware - should have included the minimal GPU (memory) required to run this model
* Figure 5 could have been made more readable
* lack of manual annotations - as mentioned the generation of the GT is great, but they should have created a small set of manually annotated cases to compare NeuroNet against its 'parents' to see if it learned to generalise and improve upon them

* discussion of failure cases is clear (i.e. scans failed with the original segmentation method), but ideally they would have established numbers of how many failure cases persisted with NeuroNet and how many it learned to correct

The authors should describe the variation in scanners in UK Biobank data set better. Isn't the fact that all subjects are asymptomatic a limitation? If all these healthy subjects were scanned on only a few scanners with highly standardized protocols as is common in single research studies, would NeuroNet work on more diverse data, e.g. from clinical routine?

**Special Issue:**

Yes

---

### Comment · ~Bram_van_Ginneken1 · 2018-05-18
**Selection for longlist for special issue Medical Image Analysis**

Dear authors,

Congratulations on your acceptance to MIDL! We have selected your paper on the longlist for the Medical Image Analysis Special Issue. Please read this page:
https://midl.amsterdam/special-issue-in-medical-image-analysis/
Please answer the three questions that are listed on that page about your interest in submitting to the special issue, potential overlap with other publications, and related publications.

You can post your answer here directly below on openreview.net, or mail me directly at bram.vanginneken@radboudumc.nl.

Best regards, Bram

---

### Decision · Program_Chairs · 2018-05-15
**Paper59 Acceptance Decision**

Poster